# Human Colorectal Cancer Infrastructure Constructed by the Glycocalyx

**DOI:** 10.3390/jcm8091270

**Published:** 2019-08-22

**Authors:** Masahito Tachi, Hideshi Okada, Nobuhisa Matsuhashi, Genzou Takemura, Kodai Suzuki, Hirotsugu Fukuda, Ayumi Niwa, Takuji Tanaka, Hideki Mori, Akira Hara, Kazuhiro Yoshida, Shinji Ogura, Hiroyuki Tomita

**Affiliations:** 1Department of Emergency and Disaster Medicine, Gifu University Graduate School of Medicine, Gifu 501-1194, Japan; 2Department of Surgical Oncology, Gifu University Graduate School of Medicine, Gifu 501-1194, Japan; 3Department of Internal Medicine, Asahi University School of Dentistry, Mizuho 500-8523, Japan; 4Department of Tumor Pathology, Gifu University Graduate School of Medicine, Gifu 501-1194, Japan; 5Department of Diagnostic Pathology (DDP) and Research Center of Diagnostic Pathology (RC-DiP), Gifu Municipal Hospital, Gifu 501-1194, Japan

**Keywords:** colorectal cancer, glycocalyx, electron microscopy, vasculogenic mimicry, angiogenesis

## Abstract

Cancer cells can survive and grow via angiogenesis. An alternative but controversial theory is cancer cells may grow via vasculogenic mimicry (VM), in which the cancer cells themselves construct vessel-like channels that are considered a leading cause of drug resistance. The dynamic functions of the glycocalyx (GCX), a meshwork composed of proteoglycans and glycoproteins that surrounds cell membranes, have been observed in endothelial cells within tumors. However, the actual structural shape formed by the GCX in human patients remains unclear. Here, we visualized the three-dimensional (3D) network structure constructed by bulky GCX in human colorectal cancer (CRC) patients using scanning electron microscopy with lanthanum nitrate staining. The network structure extended throughout the cancer cell nest, opening into capillaries, with a tunnel channel that exhibited a net- and spongy-like ultrastructure. The expression of endothelial and cancer-specific GCX-binding lectins was dramatically increased in the interstitial spaces between cancer cells. Even accounting for the presence of artifacts resulting from sample preparation methods, the intercellular tunnels appeared to be coated with the bulky GCX. Further, this 3D network structure was also observed in the tumors of *Apc*^Min/+^ mice. In conclusion, the bulky GCX modifies the network structure of CRCs in human and mice.

## 1. Introduction

Cancer cells survive by adapting to their changing environment, and require oxygen and nutrients supplied through blood vessels to rapidly proliferate [1]. Angiogenesis (neovascularization) is a widely accepted theory regarding the generation of such blood vessels in tumors, but another attractive theory, vasculogenic mimicry (VM) [2,3], was first experimentally demonstrated in 1999 [4]. VM is a blood supply system composed of cancer cells themselves, independent of endothelial vessels [2,3]. Extensive studies have evaluated VM [5]; however, the actual network structure in human patients remains unclear.

Hydrated gel-like layers (approximately 50–500 nm), called the glycocalyx (GCX), coat the surfaces of endothelial cells in the blood vessel lumen and are composed of glycolipids, glycoproteins, and proteoglycans [6]. Endothelial GCX has multiple functions, including mechanosensing and signal transduction, prevention of leukocyte adhesion and extravasation, modulation of shear stress on the plasma membrane, and regulation of the selective permeability barrier function of the structure [6], and is associated with many human diseases, including cancer [7,8].

The bulky cancer GCX enhances integrin activation and focal adhesion assembly [9]. Tension-mediated mechanotransduction by the GCX promotes a mesenchymal- or stem-like phenotype, which contributes to invasion in glioblastoma [10]. High-pressure interstitial flow through leaky capillaries and the GCX in cancer cells induce shear stress, contributing to metastasis [11]. Thus, the structural function of the cancer GCX is closely associated with cancer properties. However, little is known about the GCX structure in the cancer microenvironment of human patients.

The three-dimensional (3D) ultrastructure of the endothelial GCX has been dynamically visualized in murine models of human disease by scanning electron microscopy (SEM) with electron staining [12,13,14]. The surface of endothelial cells in the lumen of healthy capillaries is coated with the 3D mesh-like ultrastructure of the GCX layer, although injured capillaries have also been observed when visualizing the 3D ultrastructure of the GCX during early microvascular endothelial dysfunction [12,13,14].

We hypothesized that 3D cancer-specific structures are constructed with or coated by the GCX, supplying nutrients and oxygen to the tumor from the surrounding capillary vessels. The surface layer of colorectal epithelium is covered by the GCX in healthy conditions [15]. The surface GCX layer of colorectal cancers (CRCs) gets thin during the adenoma-carcinoma sequence [16], however, the GCX in the intracellular space of CRC cancer cells is bulky [17]. Here, we visualized a 3D network structure that was coated by and constructed with the bulky GCX, and which differed from the structure observed on normal epithelial cells.

## 2. Methods

### 2.1. Human Specimens and Mice

CRC and matched adjacent normal-appearing mucosa tissue samples were obtained from five patients (Appendix A and Appendix A) who underwent surgical resection in 2018 at Gifu University Hospital (Gifu, Japan). Informed patient consent and prior approval from the Gifu University Hospital Affiliated to Gifu University Graduate School of Medicine Ethics Committees (approval nos. 30-086 and 2018-098) were obtained before the clinical materials were used for research purposes.

*Apc*^Min/+^ mice were obtained from The Jackson Laboratory (Bar Harbor, ME, USA). All experiments were performed in accordance with the Gifu University International Animal Care and Use Committee guidelines for the use of animals. Prior approval from the Gifu University Ethics Committees (approval nos. 30-21) was obtained before the mouse materials were used for research purposes. Five mice were used in this study. All mice were maintained under specific pathogen-free conditions with isolated ventilation cages in an air-conditioned room with a normal 12:12 light–dark cycle. They were bred and maintained on a basal diet, CE-2 (CLEA Japan, Tokyo, Japan) until termination of the study. All mice were sacrificed and immediately removed the tissues after anesthesia. All experiments were carried out in accordance with the approved study plan and relevant guidelines.

### 2.2. Tissue Preparation

CRC tissue and matched adjacent normal-appearing mucosa tissue were prepared for examination immediately after removal from the patients (Appendix A). For murine experiments, mice were anesthetized, and macroscopic colon tumors and adjacent normal-appearing mucosa tissue samples were also prepared immediately after excision from the mice. A small piece of each sample was embedded in OCT compound and frozen with liquid nitrogen, while the other part was placed in 10% neutral buffered formalin. The frozen blocks were stored at −80 °C. Sections of frozen tissues (5–7 μm thick) were prepared with a cryostat and stained with hematoxylin and eosin (HE). Paraffin-embedded blocks were prepared from formalin-fixed tissue, and 4-μm-thick paraffin-embedded sections were produced. The remaining tissues were used for examination by electron microscopy. Several samples used in this study were frozen tissues that had been previously stored at −80 °C.

### 2.3. Immunohistochemistry for CD31 and Periodic Acid-Schiff (PAS) Double Staining

To avoid artifacts that result from some fixation methods, such as shrinkage and disruption of the extracellular matrix structure, we used frozen sections fixed in acetone for 5 min and rinsed in phosphate-buffered saline (PBS). Endogenous peroxidases were blocked for 10 min in 30% H_2_O_2_/MeOH and rinsed in PBS. Next, all sections were blocked with 2% bovine serum albumin (BSA) for 40 min at room temperature and incubated overnight at 4 °C with primary antibody (1:100 dilution) targeting human CD31 (M0823; DAKO, Glostrup, Demmark). Sections were incubated at room temperature with Simple Stain Max PO reagents (Nichirei, Tokyo, Japan) according to the manufacturer’s protocol. After washing, slides were incubated with diaminobenzidine-tetrahydrochloride (Sigma, MO, USA) and then with PAS for 15 min. All sections were counterstained with hematoxylin and cover-slipped.

### 2.4. Azan Staining

Azan staining was used for the assessment of connective tissue, with which cytoplasm is stained orange and collagen fibers are stained blue with phosphotungstic acid.

### 2.5. VM Quantification

CD31/PAS double-stained sections were viewed using light microscopy at a magnification of 400× and analyzed independently by two expert pathologists (A.N. and H.T.). Vessels/lined spaces stained with CD31 were defined as endothelium-dependent vessels. Channels enclosed by colorectal cells (identified by the absence of endothelial cells as confirmed by hematoxylin and eosin (HE) staining), lined by PAS-positive material, and negative for CD31 immunostaining were defined as VM. The average number of VM channels and endothelium-dependent vessels on each slide was determined in areas without necrosis in 10 randomly selected fields. The proportion of VM structures among of all vessels was calculated as the number of VMs/(the number of VMs + the number of CD31^+^ structures).

### 2.6. Immunofluorescence (IF) and Lectin Staining

Frozen sections were fixed in acetone for 5 min and rinsed in PBS. Sections were blocked with 2% BSA for 40 min at room temperature and incubated overnight at 4 °C with primary antibodies (1:100 dilution) targeting human CD31 (mouse monoclonal; M0823; DAKO), human CD34 (mouse monoclonal; M7165; DAKO), human pan-CK (AE1/AE3) (M3515; DAKO), human VEGF-A (rabbit polyclonal; sc-152; Santa Cruz Biotechnology, Santa Cruz, CA, USA), human CDH1 (mouse monoclonal; M3612; DAKO), VEGFR2 (rabbit polyclonal; 10782-1-AP; Proteintech Group, Chicago, IL, USA), SNX9 (rabbit polyclonal; 26415-1-AP; Proteintech Group, Chicago, IL, USA), and β-catenin (mouse monoclonal; 61053; BD Transduction Laboratories, Bedford, MA, USA). The following day, the sections were incubated with a goat anti-mouse IgG heavy and light chain specific (H&L) (Alexa Fluor 488) pre-adsorbed secondary antibody (ab150117; Vector Laboratories, Burlingame, CA, USA) or goat anti-rabbit IgG H&L (Alexa Fluor 488) pre-adsorbed secondary antibody (ab150081; Vector Laboratories) for 30 min, stained with DAPI, and mounted with Vecta fluorescent hard mount (Vector Laboratories) for examination.

For lectin staining, sections were treated with 2% BSA for 40 min. The sections were then incubated overnight at 4 °C with biotinylated lectins (Vector Laboratories) with different specificities. All lectins were applied at a concentration of 10 mg/mL. The next day, the sections were incubated with streptavidin-Dylght594 (Vector Laboratories) for 30 min, stained with DAPI, and mounted with Vecta fluorescent hard mount (Vector Laboratories) for examination. Images were acquired using an Olympus (Tokyo, Japan) or a Keyence BZ-X800 microscope (Tokyo, Japan).

For immunofluorescence (IF) and lectin double staining, primary antibodies and biotinylated lectins were mixed and incubated overnight at 4 °C. The next day, the sections were washed, incubated with streptavidin-Dylght594 (Vector Laboratories) and goat anti-mouse IgG H&L (Alexa Fluor 488) pre-adsorbed secondary antibody (ab150117; Vector Laboratories) or goat anti-rabbit IgG H&L (Alexa Fluor 488) pre-adsorbed secondary antibody (ab150081; Vector Laboratories) for 30 min, stained with DAPI, and mounted with Vecta fluorescent hard mount (Vector Laboratories) for examination.

### 2.7. Electron Microscopy

To detect endothelial GCX by electron microscopy, mice were anesthetized and perfused with a solution composed of 2% glutaraldehyde, 2% sucrose, 0.1 M sodium cacodylate buffer (pH 7.3), and 2% lanthanum nitrate through a cannula placed in the left ventricle. Prior to perfusion, an incision was made in the right atrial appendage, and the neck was ligated with a silk suture. In addition, a perfusion pump was used for injection at a steady rate of 1 mL/min. The tissue was then harvested and diced. Three or four pieces of tissue of approximately 1 mm^3^ in size were immersed in the perfusion solution for 2 h for fixation and then soaked overnight in a solution without glutaraldehyde prior to washing in alkaline (0.03 mol/L of sodium hydroxide) sucrose (2%) solution.

To detect endothelial GCX in human tissues using electron microscopy, the samples were diced into pieces of approximately 1 mm^3^ in size. These tissue pieces were then immersed in a solution composed of 2% glutaraldehyde, 2% sucrose, 0.1 M sodium cacodylate buffer (pH 7.3), and 2% lanthanum nitrate for 2 h, soaked overnight in a solution without glutaraldehyde, and washed in alkaline (0.03 M NaOH) sucrose (2%) solution. The specimens were then dehydrated through a graded ethanol series. The frozen fracture method for examination by SEM was then undertaken as described previously [12,13,14].

The specimens were examined using SEM (S-4800; Hitachi, Tokyo, Japan). To prepare samples for TEM, each specimen was embedded in epoxy resin. Ultrathin sections (90 nm) stained with uranyl acetate and lead citrate were then examined using TEM (HT-7700; Hitachi). For electron microscopy, 2.5% glutaraldehyde in 0.1 M phosphate buffer (pH 7.4) was used instead of perfusion buffer, as described above.

### 2.8. Quantitative Assessment of the GCX

Quantitative assessment of the area that the GCX occupies between normal epithelial cells and cancer cells was performed on six randomly chosen fields in TEM images using Olympus Cell Sens software (Tokyo, Japan). The automated threshold of the GCX included only the black areas that were stained with lanthanum [14].

### 2.9. Statistical Analysis

Statistical evaluation was performed using Mann–Whitney U tests to determine the significance of differences in the VM analysis and in the wide spaces between cells in normal and cancer microenvironments. All calculations were performed using Graph-Pad Prism statistical software (version 6.0; GraphPad Software Inc., San Diego, CA, USA) and results with *P* values of less than 0.05 were considered statistically significant.

## 3. Results

### 3.1. VM Formation Defined by CD31/PAS Staining Is Uncommon in Patient-Derived CRC Tissues

We carefully obtained small pieces of CRC tumor and normal-appearing mucosal tissues from specimens collected from chemotherapy-naïve patients with CRC. Specimens were obtained in the operating theater immediately after excision, and frozen sections were prepared to maintain cellular structures [18] (Figure 1a), because 10% neutral buffered formalin causes some artifacts such as tissue shrinkage as well as streaming artifacts from water-soluble glycogens [19]. We also fixed small pieces for scanning electron microscope (SEM) and transmission electron microscope (TEM) (Figure 1a).

The histological type of all cases was adenocarcinoma (Appendix A). Human CRC specimens were evaluated for VM formation, which is typically defined as the presence of channels lined with CD31− and PAS+ cancer cells, as determined by double staining (Figure 1b). The average number of CD31−/PAS+ channels (0.13 ± 0.34) in carcinomas was much lower than the number of CD31+/PAS+ capillaries (18.7 ± 6.99) in all CRC tissues (Figure 1b), suggesting that VM may be rare in human CRC.

### 3.2. SEM Imaging Visualizes Cancer Intercellular Tunnels Connected to Small Capillaries

A capillary is a small blood vessel composed of endothelial cells ranging from 5 to 10 µm in diameter, and is just wide enough for a red blood cell (RBC) to squeeze through [20]. Cancer cells are generally less than 10 μm in diameter and chaotically overlap. Because of this ultrafine structure, confocal laser scanning microscopy (CLSM) is not sufficient to visualize the relationship between cancer cells and capillaries, including VM formation. SEM imaging with electron staining can be used to examine the volume and morphology of endothelial GCX by comparing normal and disease capillaries, although images may not always reflect the real and true morphology [12,14].

In this study, we conducted SEM imaging to visualize the ultrastructures of patient-derived fresh CRC tissue to assess the volume and morphology of real-like and true-like cancer GCX by comparing normal tissue with cancer tissue.

Thus, we investigated the 3D network structure in human CRCs using SEM. Briefly, tissues were processed with lanthanum-containing alkaline fixative to detect the GCX [12].

We observed a 3D mesh-like ultrastructure of the endothelial GCX in the capillaries of normal-appearing mucosa of patients with CRC (Figure 2a), and a moss-like endothelial GCX was also shown by TEM with lanthanum-containing alkaline [12,21] (Figure 2b). In cancer tissues (Figure 2c), a bush-like ultrastructure of the endothelial GCX on the lumen of capillaries was observed, and the capillaries showed some loss of endothelial cells, exhibiting irregular pores (Figure 2c). Hence, tumor vessels frequently had widened intercellular spaces between endothelial cells, leading to leakage [22]. Red blood cells could not easily pass through these capillary pores (less than 1 to 2 μm) (Figure 2d). The capillary pores opened up to the cancer cell nest and connected to narrow spaces between cancer cells, while the spaces between the cancer cells showed many irregular net-like ultrastructures connecting cancer cells (Figure 2c). Some cancer cells entered the capillary lumen (Figure 2c).

These results suggest that the 3D network structure connecting CRC cancer cells to capillaries had spaces with a net-like ultrastructure, further supporting the VM theory.

### 3.3. Cancer Intercellular Tunnels Are Wider than Those of Normal Epithelium

In transverse sections of CRC cells, all cancer cells were surrounded by intercellular spaces with connecting bridges (Figure 3a). The surface morphology of the cancer cells was uneven, suggesting GCX bulkiness (Figure 3a). Cross sections of CRC cells showed a 3D tunnel channel structure with a net- and spongy-like ultrastructure (Figure 3a). The tunnel channels were extended throughout the cancer cell nest. In contrast, it was difficult to detect tunnel channels between epithelial cells in the crypts of normal-appearing mucosa in SEM images (Figure 3b).

Finally, we confirmed that the tunnel channels between the cancer cells contained the GCX. The backscattered high-energy electrons that rebounded from the sample surface indicated the presence of metals binding to lanthanum nitrate in the tissue [12,21]. The location of the backscattered electrons was consistent with the spaces between the cancer cells and the net-like ultrastructure connecting the cancer cells (Figure 3c), supporting the presence of the GCX.

### 3.4. CRC Has Intercellular Tunnels with GCX Even after Considering Sample Preparation Artifacts

Currently, we cannot prevent artifacts such as shrinkage, dehydration, and chemical degeneration resulting from the process of human tissue sample preparation for observation by any microscopic methods, i.e., light microscopy (LM), TEM, SEM, and CSLM. Formalin fixation generally shrinks mammalian tissues more than fresh-frozen preparation. In electron microscopy, the fixation method for TEM imaging generally requires more time and technical expertise than that of SEM, leading to more artifacts than SEM. Even 3D reconstruction from images of CLSM with fluorescent staining cannot avoid formalin fixation. Therefore, interpretation of microscopy requires knowledge of the fine structure of cancer and experience of the numerous artifacts that arise from fixation, embedding, sectioning, and contrast staining [23].

Intercellular tunnels in normal and cancer tissues were assessed by four certified pathologists that are employed as diagnostic pathologists at different hospitals (H.T., A.H., H.M., and T.T.). We prepared fresh-frozen and formalin-fixed sections for LM and fixed tissue for TEM and SEM (Figure 4). Surprisingly, the intercellular width of both normal epithelial and cancer cells in formalin-fixed sections for LM was slightly smaller than fresh-frozen sections, however, Azan-positive spaces, indicating the presence of collagen fibers, clearly existed in the intercellular spaces between cancer cells (Figure 4a,b).

SEM and TEM images also showed that CRC tissues had much wider intercellular tunnels than normal epithelium (Figure 4c,d). Hence, even after considering the possibility of artifacts resulting from sample preparation, wide intercellular tunnels were evident in the patient-derived CRC tissues.

Azan-staining images of both fresh-frozen and formalin-fixed sections revealed the existence of many more collagen fibers in the intercellular tunnels of CRC than those of normal epithelium (Figure 4a,b). Further, TEM imaging showed bridge-like structures between each cancer cell with many scattered GCX structures in the intercellular tunnels of CRC (Figure 4c).

The channels between cancer cells were wider than those between normal epithelial cells (Figure 4e). Thus, the 3D tunnel channel structure was constructed through the bulkiness of GCX.

Taken together, these results suggest that the wide intercellular tunnels may be maintained with collagen fibers and coated with bulky GCX in patient-derived CRC compared with normal epithelium; we are currently confirming if the obtained SEM images were real and true through ongoing experiments.

### 3.5. The Intercellular Tunnels of Cancer Tissue Contain Rich Sugar, the Component of Cancer-Specific GCX

Next, to identify whether bulky GCX contains cancer- or endothelial-specific GCX, we stained for lectins, which bind glycoproteins located in the GCX [8]. To explore the specific lectins associated with human CRCs, we performed lectin staining with endothelial-specific Ulex europaeus agglutinin I (UEA-I) [24], cancer-specific Vicia villosa lectin (VVL) [25], and others using immunofluorescent (IF) staining (Appendix A and Appendix A).

UEA-I was expressed only in the endothelial cells of normal crypts but not in epithelial cells (Figure 5a). In contrast, VVL was expressed only in epithelial cells with mucin in normal crypts (Figure 5a). UEA-I and VVL were expressed in the intercellular spaces between cancer cells in human CRC tissues (Figure 5b). UEA-I was also expressed in CD31+ and CD34+ endothelial cells; however, VVL was not expressed in either. UEA-I and VVL were colocalized in cancer membranes stained with pan-cytokeratin, AE1/AE3 (Figure 5b), and these lectins showed spreading around and outside of the cancer cell membranes. Thus, the bulky GCX in CRCs was composed of cancer- and endothelial-specific sugar proteins, suggesting that cancer GCX might have endothelial properties.

We also confirmed high expression of vascular endothelial growth factor (VEGF) in the analyzed CRC tissues by IF staining (Figure 5c), although its receptor, vascular endothelial growth factor receptor 2 (VEGFR2), did not increased in cancer cells only in endothelial cells of the stroma (Appendix A). To test if there was a reduction in cell-cell adhesion protein, we conducted immunohistochemistry (IHC) for CDH1. The result showed that CDH1 expression between cancer cells was much lower than between normal epithelial cells (Figure 5d). Recently, sorting nexin 9 (SNX9) regulates invadopodia formation and cell invasion in breast cancer cells [26,27]. SNX9 protein is highly expressed in tumor and tumor-endothelial cells of human CRC tissues [28], suggesting that SNX9 may also function in cell invasion/migration in the epithelial cells of CRCs, in addition to its novel function in angiogenesis of the endothelial cells. Thus, we stained SNX9 in human CRC tissues, and the results showed that high SNX9 expression in CRC cells (Appendix A).

### 3.6. 3D Ultrastructure of Mouse Colon Cancer Phenocopies Human CRC

The 3D network structure observed in human CRCs was structurally mimicked in the colon adenoma-carcinoma tumors of older *Apc*^Min/+^ mice (Figure 6a,b), showing multiple spontaneous tumors within the small intestine and colon [29]. The result suggests that this network might consist of universal ultrastructural morphology constructed by the GCX during tumorigenesis. Further, the *Apc*^Min/+^ colon tumor cells with β-catenin accumulation in the cytoplasm had bulky VVL lectins in the interstitial space between tumor cells but not between normal crypt cells (Figure 6c), suggesting that the increase of GCX around tumor cells might be associated with Wnt/β-catenin-related colon carcinogenesis.

## 4. Discussion

In human CRC tissues, we visualized a 3D network that was constructed with GCX and which extended throughout the cancer cell nest (Figure 7). Even accounting for artifacts that can result from sample preparation methods, the structure appeared to represent the ultimate “cancer infrastructure” and could permit nutrients to reach cancer cells from distantly located blood vessels.

Based on our findings, VM seemed to be part of this cancer infrastructure. CD31 and CD34 endothelial markers were not expressed in human CRC cells in our cases (Figure 3); however, the CRC cells exhibited a GCX-coated network structure connected to the capillaries of the tumors, suggesting that VM formation may occur in human CRCs. In fact, cancer tunnel channels constructed with the bulky GCX were unlikely to carry red blood cells from capillaries to cancer cells due to their small diameter and net-like ultrastructure. However, cancer cells can survive by switching to oxygen-independent metabolism. Furthermore, the network could supply nutrients to cancer cells and release substances into the blood circulation. Cancer cells must also supply biological factors through this cancer infrastructure via autocrine and paracrine secretion. Thus, cancer cells can survive and proliferate rapidly.

The cancer-derived 3D network coated with GCX is considered an ideal structure to promote the survival and proliferation of cancer cells. Although interstitial fluid flow within tumors is typically very slow [30], GCX consists of a hydrated gel-like layer with a large surface area and a spongy-like structure, and thus may be useful for the maintenance of water, nutrient, and substance concentrations. Furthermore, the sponge-like structure may prevent the intercellular space between cancer cells from collapsing and may also function as a shock absorber. Plus, the net-like structure may protect immune cells from infiltrating the intratumor space and encountering cancer cells. Thus, these structural characteristics may be associated with drug resistance.

We did not clarify the functions of GCX and functional significance of the wide intracellular space in cancer in this study. However, based on the appearance of the 3D network structure, cancer cells must increase the surface area of the GCX layer on which they can catch and release many substances and biological factors effectively. This supports a previous finding that the bulkiness of GCX enhances integrin-dependent signals, thereby contributing to invasion [9]. Structurally, cancer cells likely acquire their mesenchymal- or stem-like phenotypes through mechanotransduction factors, such as tension, compression, or shearing in peritumoral invasion area [10]. Furthermore, the shear stress between cancer cells in the intratumor space may also enhance changes in the phenotypes of cancer cells.

Further elucidation of the structural and functional mechanisms of the cancer infrastructure in various human cancer types will contribute to the development and improvement of anticancer therapies, particularly drugs and drug delivery systems, through the establishment of proper research tools.

## 5. Conclusions

Our work provides an evidence that the 3D cancer structure required for cancer cell survival and proliferation consists of sugar-based glycocalyx-coated cancer cells in human.

## Figures and Tables

**Figure 1 jcm-08-01270-f001:**
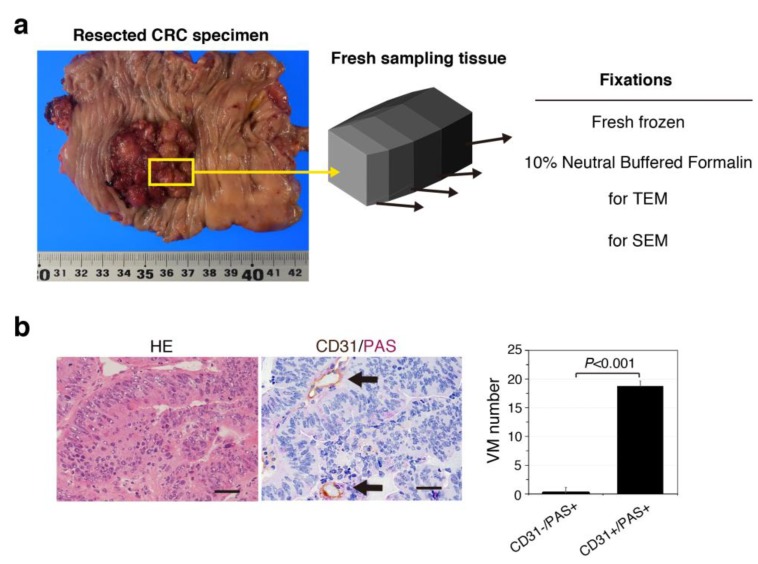
Vasculogenic mimicry (VM) formation defined by CD31−/PAS+ expression is uncommon in patient-derived colorectal cancer (CRC) tissues. (**a**) Image of a surgical specimen obtained from a human CRC patient. Cubic-shaped tissues of 10 × 5 × 5 mm were resected from normal-appearing tissue and cancerous lesions. The samples were sectioned and used accordingly. (**b**) Images of HE-staining (left) and CD31/PAS expression (right) in CRC tissues. Arrows indicate CD31+/PAS+ capillary vessels, but not VM formation. Scale bars represent 50 μm. (**b**) VM quantification in five CRC tissue samples. Data: Averages ± standard deviations.

**Figure 2 jcm-08-01270-f002:**
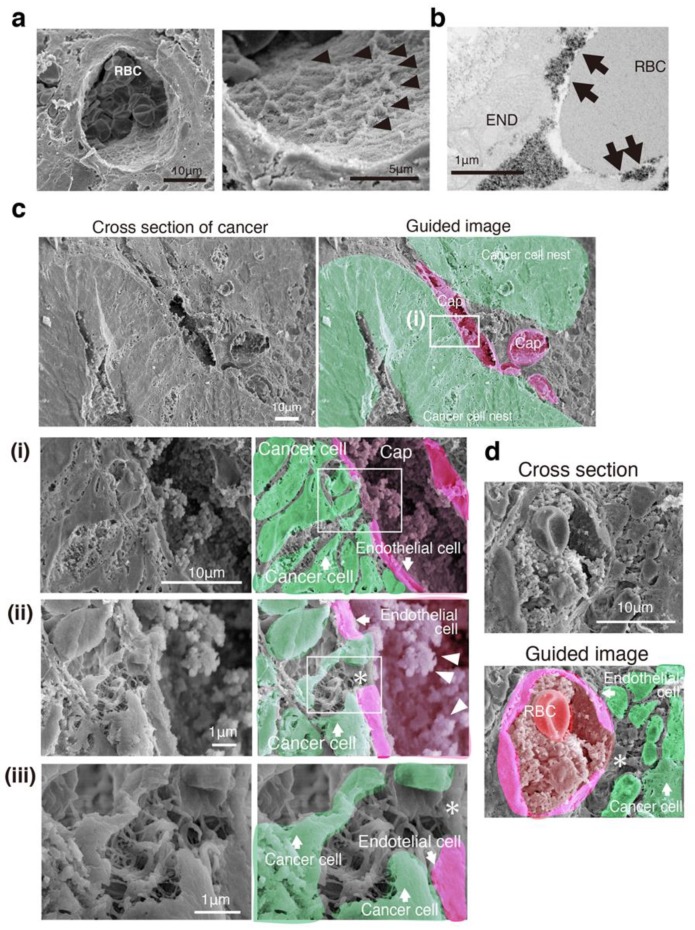
Three-dimensional network structure of a cancer capillary channel surrounded by human CRC cells. (**a**) SEM image with lanthanum nitrate of a vessel in normal-appearing mucosa. Right image: High magnification of the left image. Arrowheads indicate 3D mesh-like endothelial glycocalyx (GCX). RBC: Red blood cell. (**b**) TEM image with lanthanum nitrate of a vessel in normal-appearing mucosa. Arrows indicate moss-like endothelial GCX. RBC: Red blood cell. END: Endothelial cell. (**c**) SEM image with lanthanum nitrate (left) and guided image (right) showing the relationship between cancer cell nests and capillary vessels in a cross section. Cap: Capillary. **(i–iii)** High-magnification image (left) and guided image (right) of the rectangles in (**i–iii**), respectively. Asterisk indicates uncoated capillary wall. (**d**) A pore of the capillary opened and connected to the network structure composed of the cancer cells themselves in another patient. RBC: Red blood cell. Asterisk: A pore of the capillary.

**Figure 3 jcm-08-01270-f003:**
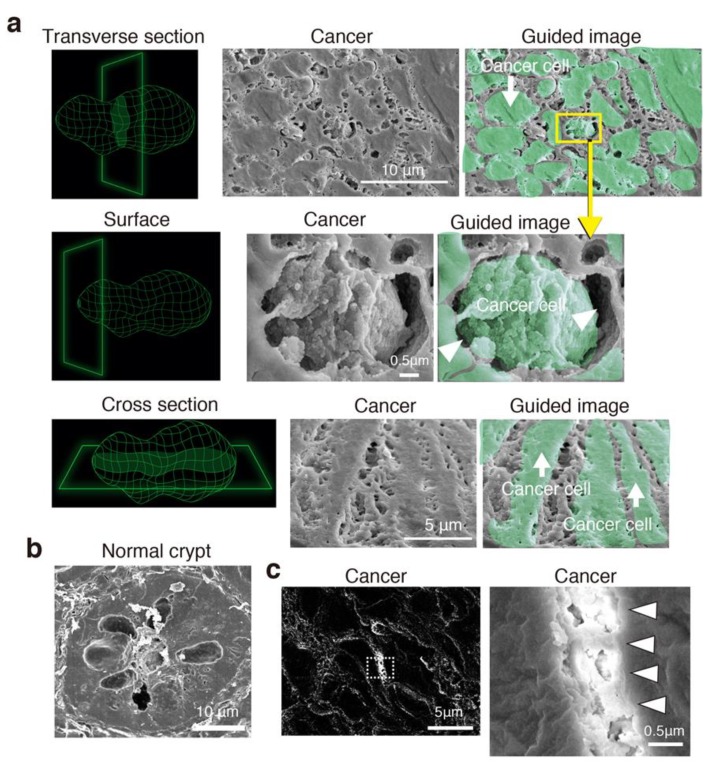
Three-dimensional structure coated by the GCX in patient-derived CRC cells. (**a**) SEM and guided images with lanthanum nitrate showing transverse, surface, and cross sections of representative cancer tissue. Left diagrams indicate a representative cancer cell. Arrowheads indicate interstitial channels with bridge-like GCX. (**b**) SEM images of transverse sections of a normal-appearing crypt. (**c**) Backscattered electron micrograph of cancer tissue. (Right) High-magnification image of the dotted-line rectangle in the left-hand image. The bush- and bridge-like structure includes lanthanum, indicating that this structure is the GCX (arrowheads).

**Figure 4 jcm-08-01270-f004:**
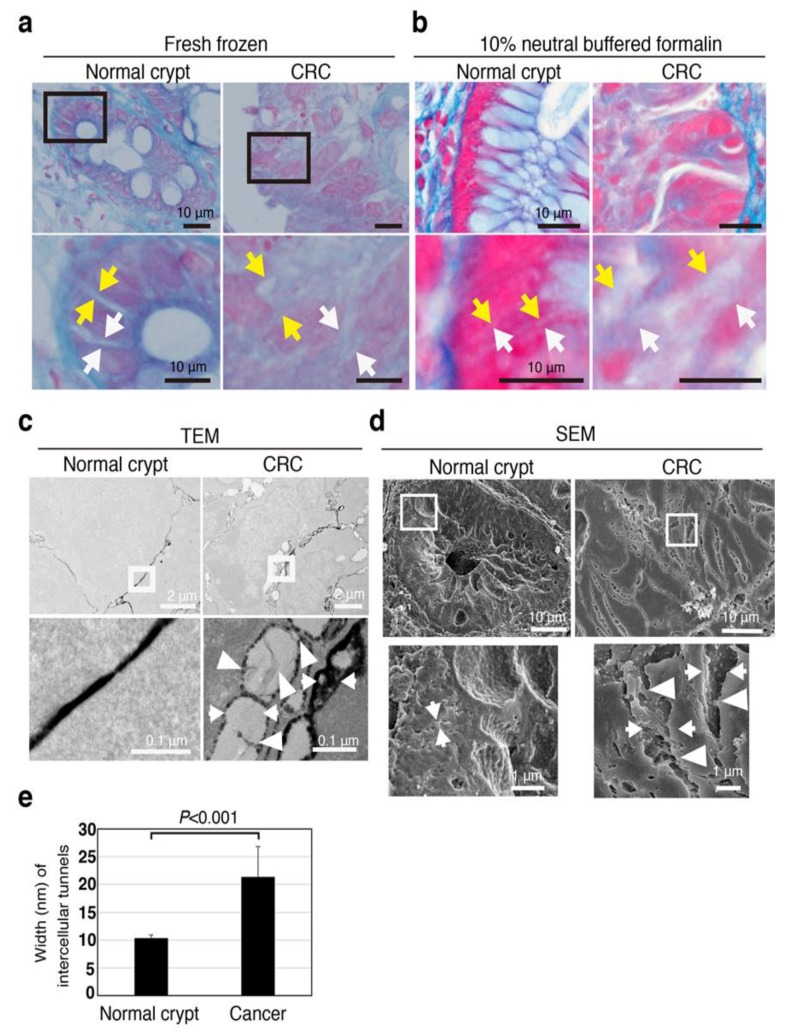
Intercellular tunnels are wide in human CRC tissues regardless of fixation procedure. (**a**) Images of Azan staining in fresh-frozen tissues (normal and cancer tissue) of a CRC patient. Arrows (both white and yellow) indicate the width of intercellular tunnels. (**b**) Images of Azan staining in 10% neutral buffered formalin-fixed tissues (normal and cancer tissue) of a CRC patient. Arrows (both white and yellow) indicate the width of intercellular tunnels. (**c**) TEM image with lanthanum nitrate of representative normal and cancer tissue. Arrows indicate the width of intercellular tunnels. Arrowheads indicate bridge-like ultrastructures. (**d**) SEM image with lanthanum nitrate of representative normal and cancer tissue. Arrows indicate the width of intercellular tunnels. Arrowheads indicate bridge-like ultrastructures. (**e**) Intercellular tunnels between cancer cells were wider than those between normal-appearing epithelial cells in TEM images with lanthanum nitrate. Error bars represent ± standard deviations of the average values.

**Figure 5 jcm-08-01270-f005:**
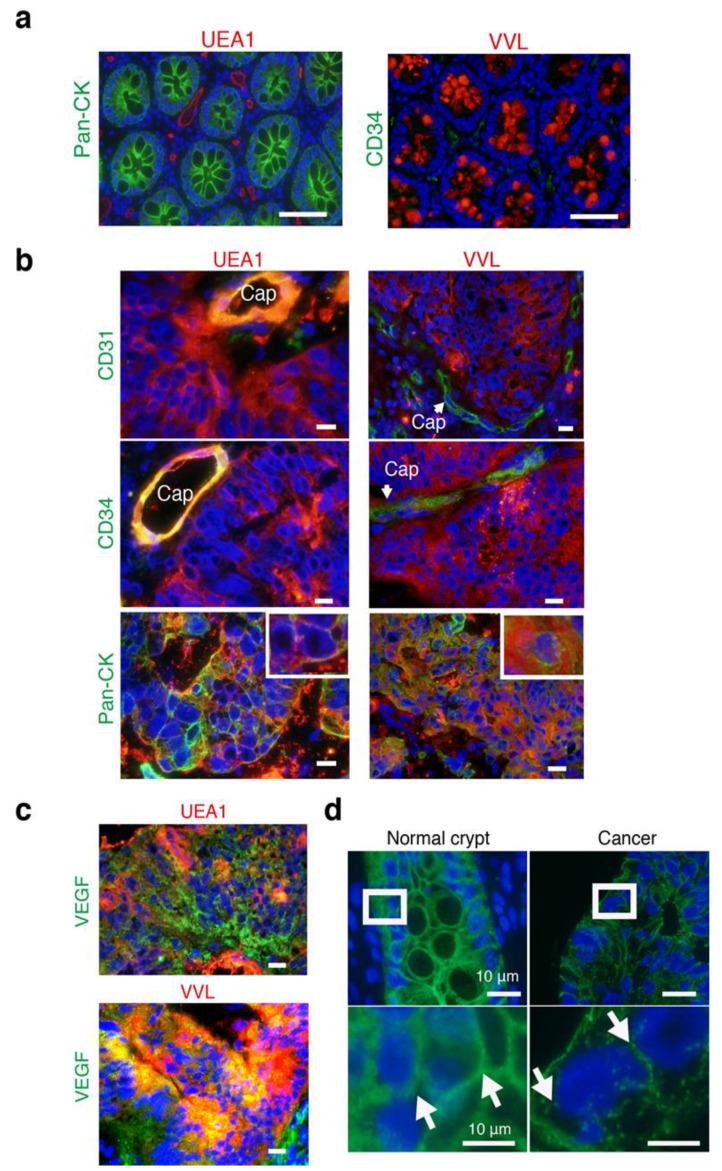
Cancer- and endothelial-related lectins expressed in cancer intercellular regions. (**a**) Merged images illustrating IF staining of lectins (UEA-1 and VVL: Red) and antibodies (pan-CK or CD34: green) with DAPI (blue). Scale bars: 10 μm. (**b**) Merged images illustrating IF staining of lectins (UEA-1 and VVL: red) and antibodies (CD31, CD34, and pan-CK: green) with DAPI (blue). Insets in the Pan-CK panel indicate high magnification. Cap: capillaries. Scale bars: 10 μm. (**c**) Merged images illustrating IF staining of lectins (UEA-1 and VVL: red) and antibodies (VEGF: green) with DAPI (blue). Scale bars: 10 μm. (**d**) Merged images illustrating immunofluorescent (IF) staining of CDH1 antibody (green) with DAPI (blue). Insets in upper images indicate lower high magnification images. Scale bars: 10 μm.

**Figure 6 jcm-08-01270-f006:**
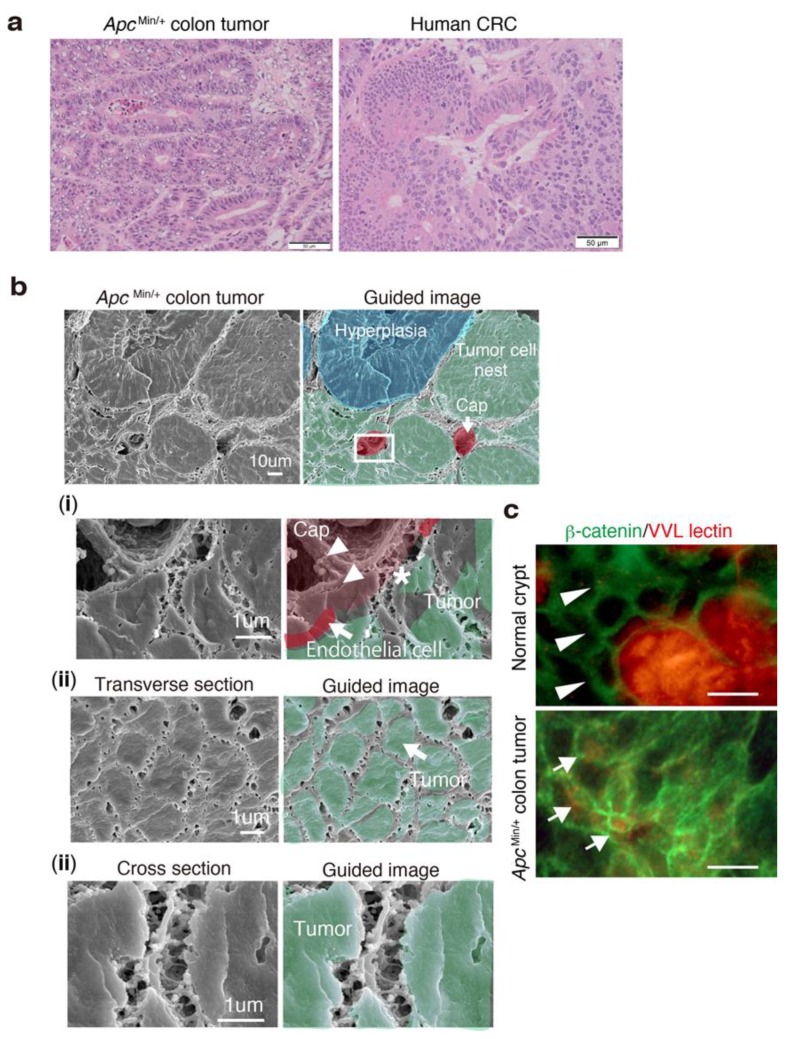
Murine colon cancer phenocopies the 3D network structure in human CRC tissues. (**a**) Representative hematoxylin and eosin (HE) images of frozen sections of *Apc*^Min/+^ colon tumor (left) and human colorectal cancer (right). Older *Apc*^Min/+^ mice developed several advanced colon tumors at 30 weeks of age, mimicking human adenocarcinomas. (**b**) Representative SEM and guided images of a colon tumor in an *Apc*^Min/+^ mouse at 30 weeks of age. (**i**) High magnification of the rectangle in the upper image showing the relationship between a tumor cell nest and a capillary. (**ii**) Transverse and (**iii**) cross sections of *Apc^Min/+^* colon tumors. (**c**) Double IF staining of β-catenin (green) and VVL lectin (red) in the normal crypts and colon tumors of older *Apc*^Min/+^ mice. Scale bars indicate 5 μm.

**Figure 7 jcm-08-01270-f007:**
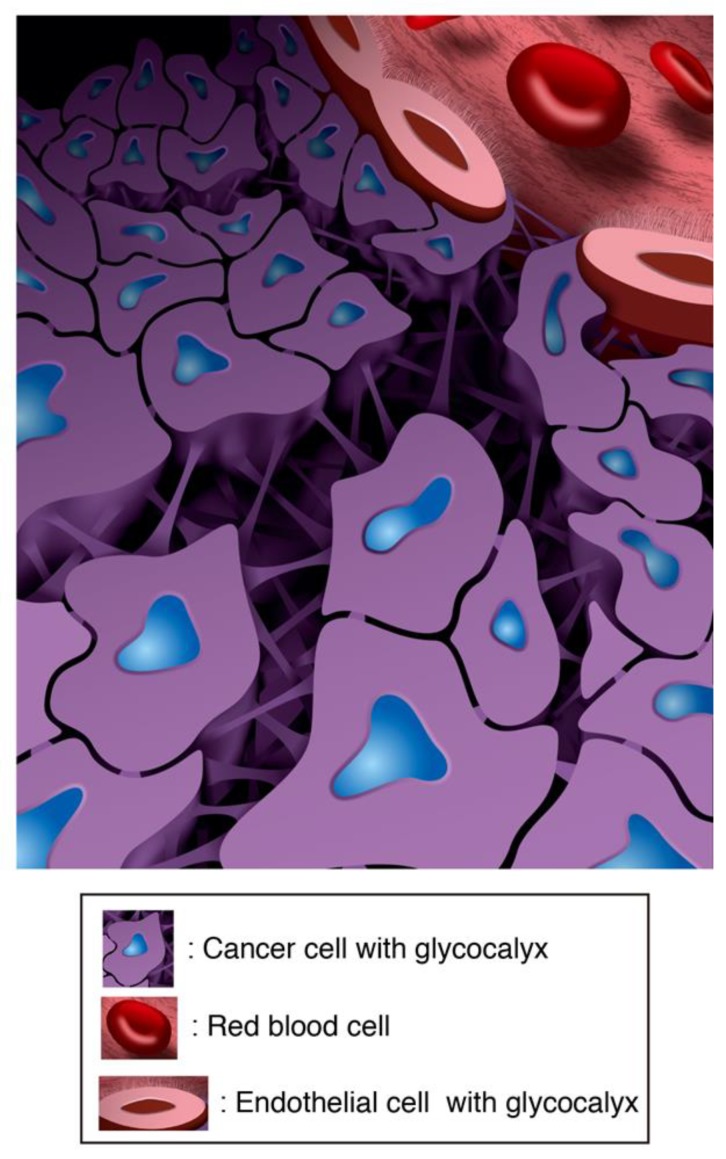
Diagram showing three-dimensional (3D) cancer infrastructure in human CRC.

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
