# Peer review of "Human Colorectal Cancer Infrastructure Constructed by the Glycocalyx"

_jcm, 2019, doi:10.3390/jcm8091270_

Round 1

Reviewer 1 Report

Authors carefully observed the infrastructures by glycocalyx in human and mouse colorectal cancers using SEM and IHC. The quality of images and the conclusions are overall good. The concept is interesting. My only concern is below, which are relatively minor.

1) In Fig. 5 and Fig. 6, authors should characterize the colorectal cancer intracellular regions in more details. A recent study showed that SNX9 is a very good marker of both tumor endothelial cells and epithelial cells in human colorectal cancers (Tanigawa et al., J. Cell. Physiol. 2019; https://www.ncbi.nlm.nih.gov/pubmed/30784076). Authors should perform IHC of human and mouse colorectal cancer tissues using the SNX9 antibody. In addition, VEGFR2 also should be stained in colorectal cancer intracellular regions. Co-staining of SNX9 and UEA1 (or VVL), VEGFR2 and UEA1 (or VVL) will improve and support the conclusions. 

Author Response

Please see the attachment or below.

-------------------------------

Response letter

To Reviewer #1

We appreciate your comments on our manuscript. Our point-by-point responses to all comments have been provided below.

Thank you.

Comments 

Authors carefully observed the infrastructures by glycocalyx in human and mouse colorectal cancers using SEM and IHC. The quality of images and the conclusions are overall good. The concept is interesting. My only concern is below, which are relatively minor.

1) In Fig. 5 and Fig. 6, authors should characterize the colorectal cancer intracellular regions in more details. A recent study showed that SNX9 is a very good marker of both tumor endothelial cells and epithelial cells in human colorectal cancers (Tanigawa et al., J. Cell. Physiol. 2019; https://www.ncbi.nlm.nih.gov/pubmed/30784076). Authors should perform IHC of human and mouse colorectal cancer tissues using the SNX9 antibody. In addition, VEGFR2 also should be stained in colorectal cancer intracellular regions. Co-staining of SNX9 and UEA1 (or VVL), VEGFR2 and UEA1 (or VVL) will improve and support the conclusions.

 Response:Thank you for your suggestions. We have co-stained the sections of normal and colorectal cancer tissues with SNX9/UEA1, VEGFR2/UEA1, SNX9/VVL, and VEGFR2/VVL.

In human tissues, all the co-stains have done very well. Briefly, high SNX9 expression was shown in colorectal cancer cells compared with normal colorectal epithelial cells. As Tanigawa et al.have described, the SNX9-positive endothelial cells were also shown in the cancer stroma. Tanigawa et al. have concluded that SNX9 may also function in loosening the cell-cell interaction in the epithelial cells of colorectal cancers, in addition to its novel function in angiogenesis of the endothelial cells. This theory supports our results.

VEGFR2 expression was seen only in endothelial cells but not in cancer tissues. This result suggest that colorectal cancer cells may differentiated into endothelial cell.

We have added these results to the Results section on lines 220-221 and 224-229. These photos have added asFig. S2. Used antibodies are listed in the Method section onlines 359-360.Figure legends of Fig. S2 have added on lines 556-560.

Finally, we have added “Tanigawa et al., J. Cell. Physiol. 2019”to our references.

Unfortunately, in mouse tissues, co-staining did not go well. Our study mainly has focused on the human tissue of colorectal cancer, and thus we did not add the mouse tissue staining.

Finally, we thank you for your valuable comments and suggestions, which have enabled us to considerably improve our manuscript.

Response letter

To Reviewer #2

 We appreciate your comments on our manuscript. Our point-by-point responses to all comments have been provided below.

Thank you.

 Comments

The authors used scanning electron microscopy to visualize glycocalyx formed three-dimensional network structure in human and mouse colorectal cancer tissues. They proposed that this glycocalyx formed three-dimensional network structure served as vasculogenic mimicry in colorectal cancer. In general, the manuscript is well-written, but it is descriptive and there is no evidence that this glycocalyx based structure is required for cancer cell proliferation and survival. On the contrary, there is a recent report showed that glycocalyx is decreased in colon adenoma and adenocarcinoma (PMID: 28033349).

 -It is confusing that the authors pointed out “VM formation is uncommon in patient-derived CRC tissues”, then what is the significance of VM in CRC development?

 -Even though the authors observed wider intercellular space in CRC than healthy tissue, the functional significance is still not cdelarb

 Minor:

 Line 79, Please correct the grammar for “Colorectal cancers frequently product mucins”.

 There are no “R: red blood cell. Cap: capillary.” In figure 3.

 Response:

Thank you for your comment.

Finally, we thank you for your valuable comments and suggestions, which have enabled us to considerably improve our manuscript.

Reviewer 2 Report

The authors used scanning electron microscopy to visualize glycocalyx formed three-dimensional network structure in human and mouse colorectal cancer tissues. They proposed that this glycocalyx formed three-dimensional network structure served as vasculogenic mimicry in colorectal cancer. In general, the manuscript is well-written, but it is descriptive and there is no evidence that this glycocalyx based structure is required for cancer cell proliferation and survival. On the contrary, there is a recent report showed that glycocalyx is decreased in colon adenoma and adenocarcinoma (PMID: 28033349).

-It is confusing that the authors pointed out “VM formation is uncommon in patient-derived CRC tissues”, then what is the significance of VM in CRC development?

-Even though the authors observed wider intercellular space in CRC than healthy tissue, the functional significance is still not cdelarb

Minor:

Line 79, Please correct the grammar for “Colorectal cancers frequently product mucins”.

There are no “R: red blood cell. Cap: capillary.” In figure 3.

Author Response

Please see the attachment or below.

--------------------------------------------

Response letter

 To Reviewer #2

 We appreciate your comments on our manuscript. Our point-by-point responses to all comments have been provided below.

Thank you.

 Comments

1) The authors used scanning electron microscopy to visualize glycocalyx formed three-dimensional network structure in human and mouse colorectal cancer tissues. They proposed that this glycocalyx formed three-dimensional network structure served as vasculogenic mimicry in colorectal cancer. In general, the manuscript is well-written, but it is descriptive andthere is no evidence that this glycocalyx based structure is required for cancer cell proliferation and survival. On the contrary, there is a recent report showed that glycocalyx is decreased in colon adenoma and adenocarcinoma (PMID: 28033349).

 Response: Yes, I agree with your underlined comments. Thank you for your excellent questions. We have responded as below.

 A)there is no evidence that this glycocalyx based structure is required for cancer cell proliferation and survival.

Response: Currently, there is no direct evidence that the glycocalyx based structure is required for cancer cell proliferation and survival. We have discussed this matter on lines 274-281. However, further studies on this matter will be needed. Our manuscript is thought to be the basis for future researches demonstrating the functional connection between glycocalyx and cancers.

 B)there is a recent report showed that glycocalyx is decreased in colon adenoma and adenocarcinoma (PMID: 28033349).

Response: This report (PMID: 28033349) (1) describes the surface glycocalyx layer of colorectal luminal epithelium decreases in adenoma and adenocarcinoma compared with normal tissues. This report has never investigated the volume of the glycocalyx in the cell-cell (intercellular) space in normal and cancer tissues.

  Hägerbäumer P et al. (Anticancer Res. 2015)(2) have reported the volume of the glycocalyx in the cell-cell (intercellular) space in cancer tissues increases compared with normal tissues by using lectin staining with eight different lectins. This report strongly supports our observation that the glycocalyx is much more bulky in cancer tissues than normal tissues.

 We have added the sentence, “The surface GCX layer of CRC is getting thin during adenoma-carcinoma sequence (16), however, the GCX in the intracellular space of CRC cancer cells is bulky (17).” into the Introduction section on lines 80-82.

 Refs)

(1) ((16) in the revised manuscript): K. Ramaker, S. Bade, N. Rockendorf, B. Meckelein, E. Vollmer, H. Schultz, G. W. Froschle, A. Frey, Absence of the Epithelial Glycocalyx As Potential Tumor Marker for the Early Detection of Colorectal Cancer. PLoS One 11, e0168801 (2016).

(2) ((17) in the revised manuscript): P. Hagerbaumer, M. Vieth, M. Anders, U. Schumacher, Lectin Histochemistry Shows WGA, PHA-L and HPA Binding Increases During Progression of Human Colorectal Cancer. Anticancer Res 35, 5333-5339 (2015).

 2) -It is confusing that the authors pointed out “VM formation is uncommon in patient-derived CRC tissues”, then what is the significance of VM in CRC development?

 Response: Thank you for your comment. Yes, it is confusing. We mean that “VM formation defined by CD31/PAS stainingis uncommon patient-derived CRC tissues”. We have described that SEM and TEM is much better than the definition of VM formation by CD31/PAS staining.

Thus, we have added the words, “ VM formation defined by CD31/PAS staining” into the Result section on line 85.  

 3) -Even though the authors observed wider intercellular space in CRC than healthy tissue, the functional significance is still not cdelarb

 Response: Yes, I agree with you. Currently, it remains still unknown in the functional significance of wider intercellular space in CRC than healthy tissue. In this point, further studies on this matter will be needed. Our manuscript is thought to be the basis for future researches demonstrating the functional connection between glycocalyx and cancers.

Thus, we have added the sentence, “We did not clarify the functions of GCX and functional significance of the wide intracellular space in cancer in this study.”into the Discussion section on lines 282-283.

 Minor:

4) Line 79, Please correct the grammar for “Colorectal cancers frequently product mucins”.

Response:Thank you so much for your comment. I entirely agree with you. Thus, we have removed this sentence and have rewrite, “The surface layer of colorectal epithelium is covered by the GCX in healthy conditions (15).”on lines 79-80.

5) There are no “R: red blood cell. Cap: capillary.” In figure 3.

Response:

Thank you for your comment. We have removed “R: red blood cell. Cap: capillary.” from Figure 3 legend on lines 159.  

Overall, we have revised references. Please see References (revised) on lines 475-553.

Finally, we thank you for your valuable comments and suggestions, which have enabled us to considerably improve our manuscript.

Round 2

Reviewer 1 Report

The manuscript has been revised very well according to reviewers' comments. The revised paper is suitable for publication.

Reviewer 2 Report

Publish as it is.